# Recommendations for Upper Limb Motor Recovery: An Overview of the UK and European Rehabilitation after Stroke Guidelines (2023)

**DOI:** 10.3390/healthcare12141433

**Published:** 2024-07-18

**Authors:** Daniel O’Flaherty, Khalid Ali

**Affiliations:** 1University Hospital Southampton NHS Foundation Trust, Hampshire SO16 6YD, UK; 2Department of Geriatrics, Brighton and Sussex Medical School, Brighton BN1 9PX, UK; 3University Hospitals Sussex NHS Foundation Trust, Princess Royal Hospital, Haywards Heath RH16 4EX, UK

**Keywords:** review, stroke, upper limb, arm, motor recovery, motor rehabilitation, guidelines

## Abstract

Upper limb impairment is common after stroke, with a significant impact on the stroke survivor’s function, social participation and quality of life. Clinical guidelines are needed to inform clinical practise, tailor interventions to improve outcomes and address unresolved research questions. This review contributes to the evidence guiding clinical practise for upper limb motor recovery after stroke by summarising the recommendations from the UK rehabilitation guidelines (National Institute for Health and Care Excellence (NICE) and the Intercollegiate Stroke Working Party (ISWP)) and the European Stroke Organisation (ESO) guidelines, all published in 2023. All three guidelines target researchers, clinical practitioners, stroke survivors and their social networks. An important feature in all three guidelines was therapeutic intensity, with all guidelines recommending increased therapeutic intensity. Unlike the ESO, the NICE and ISWP additionally include specific research recommendations. While the NICE guidelines seem more holistic and target a wider audience, the three guidelines are complimentary. We recommend that a future consensus statement should be developed in partnership between all three organisations, agreeing on scope and using the same terminology, on recommendations to improve stroke rehabilitation in the UK and Europe.

## 1. Introduction

Over 12.2 million strokes occur globally every year, resulting in more than 101 million stroke survivors. Population statistics show a 70% increase in the incidence of strokes between 1990 and 2019, which translates to an estimated doubling to more than 200 million stroke survivors by 2050 if the current trends continue. Stroke remains the second highest cause of mortality worldwide with 6.55 million deaths in 2019 alone [1]. As the largest cause of complex and chronic disability, with greater than half of all stroke patients being affected, stroke constitutes a major public health priority [2]. In the UK, around 100,000 people have a stroke every year, and approximately 1.3 million people have survived a stroke [3]. In 2017, European figures postulated that 7.06 million disability-adjusted life years have been lost due to stroke. The European continent is predicted to have a 27% increase in demand for rehabilitation and long-term care over the next 3 decades [4].

### 1.1. Pathophysiology of Motor Deficit after Stroke

Motor activity is largely controlled by the primary motor cortex, located within the precentral gyrus. This cortex is organised topographically, where the medial aspects represent the lower limbs and trunk while the more lateral aspects control upper limb and facial movements [5]. Cerebral circulation is primarily provided by the anterior, middle and posterior cerebral arteries. Upon ischaemic insult to the central nervous system (CNS), the affected region infarcts, which results in corticospinal tract (CST) disruption and reduced innervation to the muscle groups in the affected territory. Blood supply to the upper limb motor cortex is delivered by the middle cerebral artery, and hence disruption to this vascular territory results in upper limb motor impairment [6].

In haemorrhagic stroke, bleeding in the affected brain region results in the compression of neuronal tissue via haematoma formation and increased intracranial pressure, followed by secondary inflammation and oedema, blood–brain barrier disruption and hypoperfusion in the region adjacent to the haematoma [7]. Haemorrhagic stroke generally results in more severe initial upper limb motor impairment; however, patients with this type of stroke experience a more rapid recovery of symptoms, suggesting that the long-term outcomes are similar for both types of strokes [8].

### 1.2. Upper Limb Impairment after Stroke

Upper limb impairment occurs in approximately 50–80% of acute stroke survivors and is still present in around 50% of patients 6 months later during the chronic phase [9,10,11,12]. In contrast, up to 85% of survivors can walk independently within 6 months [13]. Upper limb impairments include incoordination, paresis or paralysis; reduced somatosensation, range of movement, speed or dexterity; and hypertonia (Table 1) [9,14,15,16,17,18,19,20]. Additionally, other neurological consequences such as apraxia may impede upper limb function, where the stroke survivor is unable to execute previously learned motor skills or tasks [21]. 

Persistent upper limb motor deficit is associated with more severe disability and reduced independence. As such, it needs to be promptly addressed to enable patients to continue to perform their activities of daily living (ADLs) such as washing, dressing, cooking, eating and using the toilet [22]. Reduced motor ability puts stroke survivors at risk of having to reduce their working hours or retire from employment altogether [23]. It is reported that only half of all stroke survivors retain the same working pattern one year after their stroke [24].

### 1.3. Upper Limb Impairment and Quality of Life

Upper limb impairment after stroke extends far beyond functional deficits alone; it is well documented that a patient’s quality of life (QoL) can be heavily impacted by stroke [25]. Patients are less able to sustain previous family roles such as working or caring for their children or parents. Stroke survivors are more likely to depend on their families for support, and experience reduced satisfaction in relationships with their partners [23,26,27,28]. Physical disabilities also adversely affect their career aspirations, social life or hobbies and drastically shift the dynamic onto a situation centred around completing basic ADLs [23,26,28]. Figure 1 demonstrates the issues resulting from upper limb impairment after stroke according to the International Classification of Functioning, Disability and Health (ICF) model (Figure 1) [29].

### 1.4. Management Approaches for Upper Limb Impairment

Currently, in the UK and a large proportion of Europe, stroke patients are treated in specialised stroke units with a dedicated multidisciplinary team (MDT) promoting rehabilitation [30,31]. For patients with altered sensation, motor deficits, reduced coordination or impairment of the upper limbs, physiotherapy and occupational therapy are offered to address impaired structure and function, activity limitation and restricted participation. Alongside this, many other specific interventions are considered depending on patient circumstances. 

### 1.5. Stroke Rehabilitation Guidelines

Clinical guidelines are needed to inform clinical practise, tailor interventions to improve patients’ outcomes, direct rehabilitation service commissioners to invest in evidence-based interventions and address unresolved research questions [31]. 

In the UK, the national guidelines on stroke management and rehabilitation are produced by two major organisations, namely the National Institute for Health and Care Excellence (NICE) and the Intercollegiate Stroke Working Party (ISWP) [31,32]. Acute stroke management and rehabilitation guidelines were originally published by the NICE and ISWP in 2013 and 2016, respectively. Both guidelines were updated in 2023 to incorporate new evidence in rehabilitation interventions, including those relating to upper limb motor recovery [31,32,33,34]. 

Guidelines produced by the NICE help healthcare practitioners and commissioners to provide high-quality care that is cost-effective. These guidelines are based on rigorous and independent analysis of complex evidence, developing recommendations on key areas and promoting the uptake of best practises and policies to improve clinical outcomes. The NICE produces several evidence-based clinical guidelines on many conditions, including stroke [35]. The NICE Guidelines are developed following a rigorous methodological approach. Initially, a draft scope is created, and a guideline committee is recruited. A consultation on the draft scope is held, and a final scope is then released. The published evidence is then searched and summarised. Committee members review the evidence, including health economic evidence, and agree on a draft guideline. Subsequently, a consultation exercise is undertaken with relevant stakeholders seeking feedback which informs revisions. The guideline is then published. Guidelines are updated on a regular basis according to the latest available evidence [31,36]. The NICE stroke rehabilitation guidelines use specific terms in their recommendations, such as ‘consider’, ‘offer’, ‘do not offer’ and ‘do not routinely offer’. The strength of recommendation used depends on the level of evidence for specific interventions [31].

The ISWP, set up by the Royal College of Physicians (RCP), is dedicated to producing evidence-based stroke guidelines to improve the quality of care delivered to all adult stroke survivors in the UK and Ireland. Its committee consists of a group of senior representatives from professional bodies in England, Wales and Northern Ireland involved in stroke care (such as occupational therapists, physiotherapists and speech and language therapists), in addition to representatives from the voluntary sector and patient and carer representatives. The ISWP guidance is produced by a development group including members from the Scottish Intercollegiate Guidelines Network (SIGN) and the Irish National Clinical Programme for Stroke. The group is supported by a project team from the Sentinel Stroke National Audit Programme at King’s College London. The ISWP used a standardised seven-step methodology to develop their guidelines, including developing a scope, performing a literature search, selecting the included studies, evidence analysis, drafting recommendations, considering health economic implications, external peer review and public consultation of drafted guidelines. In the guidelines, the ISWP uses terms such as ‘should be’, (offered, considered) ‘may be’, (considered) ‘should not be’ (treated, used) and ‘should not be routinely offered’ [32]. 

The European Stroke Organisation (ESO), a pan-European society focussing on stroke-related research, also published a clinical guideline on stroke rehabilitation in 2023, including motor recovery of the upper limbs. These recommendations are based on high-quality clinical guidelines published between 2016 and 2022 from the UK, Australia/New Zealand, Canada, the United States of America and the Netherlands. An expert European working group searched the literature for national clinical practise. Stroke guidelines written in English or Dutch with a section on rehabilitation after stroke were included. Two researchers then independently reviewed the guidelines and collated interventions which were recommended by at least three of the guidelines. Since the recommendations were based on support from at least three guidelines, they had high levels of evidence and were, therefore, all ‘strongly recommended’ [37].

### 1.6. Aims and Objectives

The objective of this review was to provide a summary document of the NICE, ISWP and ESO 2023 guidelines relating to upper limb motor recovery. Our aim was to collate recommendations from all three guidelines in one review to support a future consensus statement for UK and European stroke service providers and patients. A comprehensive understanding of which interventions are recommended, at what dose, when, who delivers them and in which setting is critical to improving patients’ outcomes. The ESO 2023 guidelines were included because they complement the UK guidelines and provide a comprehensive summary of several other guidelines from Australia, New Zealand, Canada, the Netherlands and the US published between 2016 and 2022. It is worth noting that the only UK guideline referenced by the ESO in 2023 was the ISWP guidance published in 2016 [37]. NICE guidelines were not included as they dated back to 2013 [33]. Since 2016, both ISWP and NICE updated their clinical guidelines on stroke rehabilitation with new recommendations based on updated research evidence [31,32]. 

While stroke guidelines were also published previously by SIGN in 2010, they were not included in this review, because the guidelines were based on evidence which has since been updated [38]. Additionally, the SIGN guidelines have been superseded by the National Clinical Guideline for Stroke (2023), which SIGN collaborated with ISWP and other bodies to produce [32]. Similarly, other relevant guidelines such as from the US American Heart Association/American Stroke Association were not included as they were out of date due to being published in 2016. However, they were featured in the ESO 2023 guidelines to inform their recommendations [37,39]. 

## 2. Materials and Methods

The stroke rehabilitation guidelines from the UK, namely NICE and ISWP published in 2023, and the 2023 ESO guidelines were retrieved. The two reviewers individually identified sections where the terms motor, recovery, upper limb, rehabilitation, interventions, approaches or technology in relation to the upper limb were used. After discussion, both authors agreed on specific categories to be included in this review. Both authors then agreed to collate the recommendations into 3 sections and tables: general rehabilitation approaches, technology- and device-assisted rehabilitation, and spasticity management (Table 2, Table 3 and Table 4) [31,32,37]. The relevant sections on upper limb motor recovery were then analysed by the two authors independently to identify similarities and differences. Agreement between guidelines was noted if the guidelines used the terms ‘recommend’, ‘strongly recommend’ or ‘offer’. A difference between guidelines was observed when they used different terms such as ‘consider’ when the other guideline used ‘offer’ or ‘strongly recommend’. Where there was a difference in opinion between the two authors, this difference was resolved by an in-depth discussion. The result tables were then populated and revised, and a consensus was reached. The authors followed the same approach used by the ESO to inform our methodology [37].

## 3. Results

### 3.1. Upper Limb Stroke Guidelines

In this section, the two authors summarise the UK (NICE and ISWP) and ESO guidelines relating to upper limb motor recovery after stroke in Table 2, Table 3 and Table 4. Table 2 describes the general approaches to upper limb rehabilitation after stroke, while Table 3 summarises the technology- and device-assisted rehabilitation method guidelines. Similarly, Table 4 details a summary of spasticity management in the context of upper limb motor recovery (Table 2, Table 3 and Table 4) [31,32,37]. 

### 3.2. General Approaches to Upper Limb Rehabilitation after Stroke

#### 3.2.1. Therapeutic Intensity

The ISWP and NICE recommendations are the same for intensity as they both advise a minimum of three hours of therapy on at least five days per week. Personalised rehabilitation types and intensity are also recommended by NICE and ISWP, which should be offered by a range of multidisciplinary team members [31,32]. Similarly, the ESO strongly recommends that rehabilitation is structured to provide as much therapy (occupational and physiotherapy) as possible [37]. However, it does not include a specific minimum or maximum target of therapy duration or frequency to aim for in the context of upper extremity rehabilitation [37].

#### 3.2.2. Strength Training

The ISWP recommends and the ESO strongly recommends offering strength training to stroke patients, while the NICE states that strength training should be considered. All three guidelines provide differing details on specific approaches to strength training. While the ESO strongly recommends resistance training as a form of strength training, the NICE also mentions resistance training as well as bodyweight activities and weights as two other forms. Similarly, the ISWP recommends two forms of strength training: bodyweight activities and mixed training, which includes resistance training [31,32,37].

#### 3.2.3. Task-Specific Training/Repetitive Task Training

Agreement was found between all three guidelines for this approach. The NICE and ISWP recommend, while the ESO strongly recommends, offering task-specific training (TST)/repetitive task training (RTT) [31,32,37]. In terms of RTT, the ISWP recommends that RTT should be the principal rehabilitation method alongside traditional exercise and advises healthcare providers that a unilateral or bilateral upper limb approach could be taken when offering RTT, depending on the task and level of impairment. Moreover, the ISWP uniquely clarifies that cognitive impairment should not prevent patients from being offered RTT and suggests it can be enhanced with trunk restraint or priming techniques [32]. While the ISWP specifies RTT should be intensive, task-specific, functional and consisting of a high number of repetitions, the ESO advises RTT should be intensive, task-specific, progressively adaptive, goal-oriented and meaningful [32,37]. Finally, the NICE offers multiple examples of RTT to consider, such as reaching or grasping [31].

#### 3.2.4. Mental Practice

The ISWP recommends, and the ESO strongly recommends, mental practise (MP) as an adjunct to conventional upper extremity motor rehabilitation therapies [32,37]. The ISWP additionally specifies that mental practise should be offered to individuals who are both motivated and able to participate [32]. 

### 3.3. Technology- and Device-Assisted Upper Limb Rehabilitation after Stroke

#### 3.3.1. Constraint-Induced Movement Therapy

Agreement exists between the NICE and ISWP guidelines, which both recommend considering constraint-induced movement therapy (CIMT) use. The ESO guidelines differ as they strongly recommend offering traditional or modified CIMT. However, the ESO does not advise on any qualifying criteria for CIMT in terms of range of movement [31,32,37]. The NICE specifically highlights awareness of the potentially adverse effects associated with CIMT, such as low mood [31]. 

#### 3.3.2. Splinting/Orthoses

Both the NICE and ISWP also recommend that splints are fitted and monitored by appropriately trained staff [31,32].

#### 3.3.3. Electrical Stimulation

The NICE advises that electrical stimulation (ES) should continually be offered if there is evidence of functional progression demonstrated, with the ISWP similarly stating that stimulation protocols should be tailored to the individual. Meanwhile, the ISWP limits consideration of ES to wrist and finger extensors limiting function as the desired criteria. The ISWP additionally states that training related to ES should be offered to the patient, their family or carers and involved healthcare professionals [31,32]. In terms of specific forms of ES, the ESO only recommends NMES, while the ISWP specifically recommends FES and the NICE guidelines do not specify a form of ES to be offered [31,32,37].

#### 3.3.4. Vagus Nerve Stimulation

Only the ISWP guidelines include VNS, whereby they specify that transcutaneous VNS could be considered for mild-to-moderate arm weakness. Furthermore, they highlight that implanted VNS could be considered in the context of clinical trials [32]. 

#### 3.3.5. Mirror Therapy

The NICE states that mirror therapy should be started within the first 6 months following a stroke, and describe the ideal length, frequency and level of supervision for sessions [31]. 

#### 3.3.6. Robotic Therapy

The ISWP advises that robotic therapy should ideally be considered as part of a clinical trial [32].

#### 3.3.7. Virtual Reality/Telerehabilitation

Virtual reality (VR) is not specifically mentioned by the NICE or ISWP in the context of upper limb therapy, but both guidelines recommend considering telerehabilitation, an umbrella definition for rehabilitation therapies involving technology such as virtual reality devices [31,32].

#### 3.3.8. Other Approaches and Therapeutic Interventions

No advice was offered from the NICE, ISWP or ESO about range-of-motion exercises or biofeedback, or therapies such as acupuncture or non-invasive brain stimulation, in the context of upper limb rehabilitation [31,32,37]. 

### 3.4. Management of Spasticity in the Upper Limb after Stroke

#### 3.4.1. General Principles and Splinting

The NICE and ISWP encourage creating personalised goals for managing spasticity. Both bodies also suggest considering offering and monitoring simple measures to improve spasticity, including arm positioning and passive or active limb movement [31,32]. 

#### 3.4.2. Drug Therapies

Both the NICE and ISWP guidelines state that botulinum toxin injection response should be monitored and either stopped or continued according to treatment response. Of note, the ISWP also recommends that botulinum should be offered by a specialist multidisciplinary team as an adjunctive therapy alongside usual rehabilitation therapy and/or splinting or casting. The NICE and ISWP additionally recommend monitoring for potential adverse effects of oral anti-spasmodic agents such as baclofen or tizanidine. Furthermore, the ISWP guidelines uniquely mention intrathecal baclofen or intraneural phenol but state they should only be administered by a specialist MDT spasticity service. From a competence perspective, the ISWP and NICE recommend a specialist MDT approach to the assessment, treatment and follow-up of spasticity when using drug therapies [31,32].

## 4. Discussion

In one document, this review summarises three areas in relation to general approaches to upper limb rehabilitation, technology- and device-assisted rehabilitation, and spasticity management from the NICE, ISWP and ESO guidelines published in 2023.

Collectively, there are only two areas with complete agreement between all three guidelines, namely therapeutic intensity and RTT/TST [31,32,37]. However, the UK NICE and ISWP guidelines were found to have a large degree of overlap, such as on recommendations for CIMT, splinting, ES, mirror therapy and VR.

The ISWP and NICE also encourage individualised therapy approaches according to patient needs and available resources and promote the collaboration of a wide range of appropriately trained MDT members to aid stroke survivors in maximising control of pain, improving function and easing care of the upper limbs. The NICE and ISWP also provide information which is setting-specific in relation to the organisation of stroke services and the MDT membership to help clarify where stroke rehabilitation should occur, and which MDT members should be involved [31,32]. In terms of the target audience, all three guidelines are aimed at researchers, clinical practitioners, stroke survivors and their social networks, with the unified goal of improving stroke care rehabilitation. 

Differences between guidelines are inevitable due to the different approaches and methodologies undertaken by their respective committees or working groups. All three guidelines considered were published in 2023, with evidence being considered between 2016 and 2022 for the ESO, while new evidence was reviewed between 2013 and 2023 for the NICE and 2015 and 2023 for the ISWP [31,32,37]. 

Apart from clinical guidance, the ISWP and NICE also made research recommendations to help inform future upper limb rehabilitation after stroke that is evidence-based [31,40]. The NICE makes seven recommendations pertaining to the upper limb, including exploring the clinical and cost-effectiveness of increasing rehabilitation intensity from five to seven days, as well as the interventions to help manage shoulder pain as key research recommendations. Additionally, the NICE recommends identifying the groups of people that would benefit from mirror therapy and researching the clinical and cost-effectiveness of using diagnostic assessment to inform shoulder pain management and exploring the use of acupuncture, botulinum toxin A and electrotherapy for managing spasticity [31]. Similarly, the ISWP makes nine research recommendations relating to upper limb recovery after stroke. These include the effectiveness of functional ES, robotics, mirror therapy, MP, CIMT, RTT, exercise, VNS and intensive arm recovery programmes [32]. 

In contrast, the ESO 2023 guidelines do not make any research recommendations about upper limb recovery [37]. However, the ESO previously produced a 2018–2030 Action Plan for Stroke in Europe, which defined 72 research priorities within seven domains, including broad recommendations such as exploring if post-stroke upper limb problems can be more effectively managed [41]. 

Both the ESO and NICE guidelines produce a visual summary to help illustrate the stroke rehabilitation journey, to be used by stroke survivors, their families and social networks and healthcare professionals [37,42]. Specifically, the NICE has produced a summary involving key questions around stroke rehabilitation, focussing on clinical reviews, therapeutic options, healthcare staff involved in care and assistance with ADLs and returning to the community [42]. The ESO also includes a visual representation of the rehabilitation process in Figure 3 of the guidelines, which includes a cycle of setting goals, considering patient factors, following clinical guidelines, adjusting rehabilitation plans accordingly and using the recommended assessment tools. Unlike the NICE, the ESO visual summary is tailored more towards healthcare professionals and researchers, rather than patients or their social networks [37]. 

The ISWP has not produced a visual summary. However, it has published a plain-language summary of the national clinical guidelines for stroke, which incorporates key recommendations from the 2023 guideline without jargon, aimed at patients and their social networks [32,43]. 

Overall, we consider the NICE to be the most informative guideline because it has a wider target audience, includes resources for service providers and commissioners to assist with incorporating guidelines into clinical practise and provides health economic evaluation to ensure that therapeutic recommendations provide good value for money. Additionally, the NICE provides resources aimed at the general public to improve understanding of stroke management and rehabilitation for patients and their social networks. Despite this, the NICE does not address certain therapeutic interventions mentioned by the ISWP or ESO, namely VNS or MP.

Going forward, a consensus between national and international guidelines is required to help establish the most effective therapies and approaches, identify areas of practise which require more research and subsequently make definitive treatment recommendations aiming to improve patient outcomes [37]. 

This review is unique in the context of upper limb stroke rehabilitation, as it collates into one document and compares the recommendations of the ESO, ISWP, and NICE. In addition, this review summarised the clinical recommendations under three distinct categories: approaches, interventions and management of spasticity. The purpose of doing so was to signpost practitioners into relevant areas of stroke that can be more easily identified.

However, one of the key limitations is that this review focussed exclusively on guidelines produced by Western nations that were available in English, meaning that guidelines from other regions or published in other languages were not included. It is worth mentioning that one of the authors was a committee member of both NICE and ISWP. Additionally, no third author was invited to review the guidelines, which may have been helpful such as in cases of disagreement between authors. Finally, Table 4 on spasticity management is not inclusive of the individual types of neurotoxins recommended by the guidelines. However, more details about these recommendations can be found in their respective guideline documents [31,32,37].

## 5. Conclusions

Upper limb impairment is common after stroke, with significant impact on the stroke survivor’s function, social participation, and quality of life. This review provides a comprehensive summary of the NICE, ISWP and ESO guidelines published in 2023 for upper limb motor recovery. We believe that while the NICE is more holistic and targets a wider audience, the three guidelines are complimentary. We recommend that a future consensus statement should be developed in partnership between all three organisations, agreeing on scope and using the same terminology, on recommendations to improve stroke rehabilitation in the UK and Europe.

## Figures and Tables

**Figure 1 healthcare-12-01433-f001:**
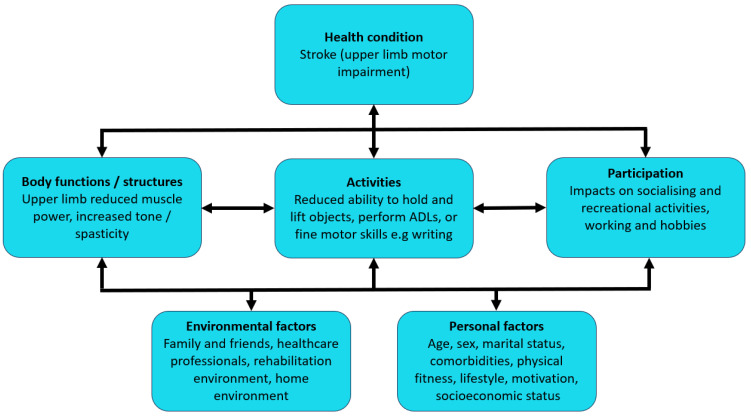
Impact of upper limb impairment after stroke according to the International Classification of Functioning, Disability and Health (ICF) model. Abbreviations: activities of daily living (ADLs) [29].

**Table 1 healthcare-12-01433-t001:** Common types of impairments or deficits of the upper limb that can occur after stroke, and associated features [9,14,15,16,17,18,19,20].

Impairment	Features
Incoordination	Inability to execute motor responses with appropriate accuracy, speed and smoothness
Paresis	Partial loss of skeletal muscle strength, reducing voluntary movement ability in the affected region
Paralysis	Complete loss of skeletal muscle strength; voluntary movement will be absent in the affected region
Somatosensory loss	Reduced sensation in modalities such as light touch, pressure, sharp–dull discrimination, temperature, pain, two-point discrimination and proprioception
Reduced range of movement	Decreased extent to which a limb or body part can be rotated around a joint
Reduced dexterity	Diminished ability to perform fine motor skills to achieve a task in an accurate and timely manner, while adapting to any environmental changes that occur
Hypertonia/spasticity	Increased muscular tone; resistance to passive movement may or may not be velocity-dependent

**Table 2 healthcare-12-01433-t002:** Summary of general approaches to upper limb rehabilitation from the National Institute for Health and Care Excellence (NICE), the Intercollegiate Stroke Working Party (ISWP) and the European Stroke Organisation (ESO) guidelines published in 2023. Abbreviations: task-specific training (TST)/repetitive task training (RTT) and mental practise (MP) [31,32,37].

General Approaches to Upper Limb Rehabilitation after Stroke	NICE	ISWP	ESO
Therapeutic intensity	Offer a minimum of 3 h of therapy on at least 5 days every week	Stroke survivors should receive a minimum of 3 h of therapy on at least 5 days of every week	Strongly recommends as much therapy as possible
Strength training	Consider strength training	Offer strength training	Strongly recommends strength training
Task-specific training/repetitive task training (TST/RTT)	Offer RTT/TST	Offer RTT/TST	Strongly recommends RTT/TST
Mental practice (MP)	No recommendations are made	Offer MP as an adjunct to conventional therapy	Strongly recommends MP as an adjunct to conventional therapy

**Table 3 healthcare-12-01433-t003:** Summary of technology-/device-assisted upper limb rehabilitation from the National Institute for Health and Care Excellence (NICE), the Intercollegiate Stroke Working Party (ISWP) and the European Stroke Organisation (ESO) guidelines published in 2023. Abbreviations: constraint-induced movement therapy (CIMT), electrical stimulation (ES), vagus nerve stimulation (VNS) and virtual reality (VR) [31,32,37].

Technology-/Device-Assisted Rehabilitation	NICE	ISWP	ESO
Constraint-induced movement therapy (CIMT)	Consider CIMT in individuals with at least 20 degrees of wrist extension and 10 degrees of finger extension	Consider CIMT in individuals with at least 20 degrees of wrist extension and 10 degrees of finger extension	Strongly recommends traditional or modified CIMT
Splinting/orthoses	Do not routinely offer hand and wrist splints Consider hand or wrist splints for people at risk—such as immobility due to weakness or increased tone	Hand and wrist splints should not be routinely offered Splints should only be offered following an individual assessment where increased tone is found to impact on the passive or active range of movement	No recommendations are made
Electrical stimulation (ES)	Consider ES as an adjunct to conventional therapy for patients who cannot move their arm against resistance but where muscle contraction is evident	Consider ES as an adjunct to conventional therapy	Strongly recommends ES as an adjunct to conventional therapy
Vagus nerve stimulation (VNS)	No recommendations are made	Consider VNS	No recommendations are made
Mirror therapy	Consider mirror therapy as an adjunct to conventional therapy	Consider mirror therapy as an adjunct to conventional therapy	No recommendations are made
Robotic therapy	Do not offer robotic therapy	Consider robotic therapy as an adjunct to conventional therapy	Strongly recommends robotic therapy as an adjunct to conventional therapy
Virtual reality (VR)/telerehabilitation	Does not mention VR in the context of upper limb therapy	Does not mention VR in the context of upper limb therapy but does recommend considering telerehabilitation	No recommendations are made

**Table 4 healthcare-12-01433-t004:** Summary of spasticity management recommendations by the National Institute for Health and Care Excellence (NICE), the Intercollegiate Stroke Working Party (ISWP) and the European Stroke Organisation (ESO) guidelines. Abbreviations: intramuscular (IM), electrical stimulation (ES), neuromuscular electrical stimulation (NMES), functional electrical stimulation (FES), transcutaneous electrical nerve stimulation (TENS) and multidisciplinary team [31,32,37].

Spasticity Management	NICE	ISWP	ESO
General recommendations and splinting	Consider a goal-directed plan for spasticity management Consider splints when needed for spasticity	Splinting should not be routinely offered to counteract spasticity Only offer splinting or casting for spasticity following an individualised assessment	No recommendations are made
Drug therapies	Consider IM botulinum toxin for focal spasticity up to every 3 months Consider oral baclofen for generalised spasticity	Offer IM botulinum toxin for focal spasticity every 3–4 months Oral baclofen or tizanidine should be offered for generalised spasticity	No recommendations are made
Electrical stimulation (ES)	Consider a trial of ES for focal spasticity (FES, TENS or NMES)	ES should not be used to reduce spasticity Consider ES for those with wrist or finger spasticity as an adjunct to botulinum toxin injections	No recommendations are made

## Data Availability

All research data were presented in this paper.

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
