# Peer review of "Recommendations for Upper Limb Motor Recovery: An Overview of the UK and European Rehabilitation after Stroke Guidelines (2023)"

_healthcare, 2024, doi:10.3390/healthcare12141433_

Round 1

Reviewer 1 Report

Comments and Suggestions for Authors

Introduction

Clarify why only the NICE, ISWP, and ESO guidelines were selected, and discuss the potential implications of excluding other relevant guidelines.

In brief, describe additional approaches taken by NICE, ISWP, and ESO to the methodology for creating their guidelines, to give context for this manuscript on comparisons.

Methods

Clearly state the criteria used to decide on the recommendations that were included in the comparison. The latter will allow the solidity of the methodology to be established and will capture the reproducible form.

Results

If Practical - Quantitative Analysis-increase the use of quantitative data to prove how useful the different suggestions are.

Emphasize Clinical Relevance: Further, describe the clinical implications of the results and how these differences in guidelines could impact patient care and outcomes.

Discussion

I would like the authors to discuss the challenges available in different healthcare settings in implementing the guidelines while addressing the problems related to the resources available, staff training, and patient adherence.

Further, a more detailed discussion is warranted about the discrepancies between guidelines and practical implications.

Areas for Further Investigation: Provide a list of areas where additional research is required, particularly if they are items with disagreement in the guidelines. This will inform future studies and potential guideline revision.

Conclusion

Reinforce the argument in the conclusion and the fact that unifying the guidelines could lead to enhanced patient care and outcomes

Provide proposed recommendations on harmonizing the guidelines or covering any gaps identified, developing a more consistent upper limb rehabilitation post-stroke approach.

 Conflict of Interest

Independent review: Given Dr. Ali's involvement working with NICE and ISWP committees, there is a potential bias, and the use of independent reviewers for future work will add a balanced view and credibility.

Author Response

Thank you very much for your feedback. Please see the authors' responses below:

Introduction

  • Clarify why only the NICE, ISWP, and ESO guidelines were selected, and discuss the potential implications of excluding other relevant guidelines.
    • The authors amended the aims and methods sections (page 7) to explain the reasoning for only including these three guidelines and why SIGN and AHA/ASA guidelines were not included.
  • In brief, describe additional approaches taken by NICE, ISWP, and ESO to the methodology for creating their guidelines, to give context for this manuscript on comparisons.
    • Individual methodologies undertaken by NICE, ISWP and ESO have now been provided in detail in introduction section (‘stroke rehabilitation guidelines’) on pages 5/6

Materials & Methods:

  • Clearly state the criteria used to decide on the recommendations that were included in the comparison. The latter will allow the solidity of the methodology to be established and will capture the reproducible form.
    • The methodology section has been greatly expanded to explain the criteria used (pages 8/9).

Results:

  • If Practical - Quantitative Analysis-increase the use of quantitative data to prove how useful the different suggestions are
    • We have explained two categories have agreement and that there is more agreement between NICE and ISWP
    • Emphasize Clinical Relevance: Further, describe the clinical implications of the results and how these differences in guidelines could impact patient care and outcomes.
      • We have now explained why we feel NICE is the best guidelines in the discussion section as well as the implications of the review.

Discussion:

  • I would like the authors to discuss the challenges available in different healthcare settings in implementing the guidelines while addressing the problems related to the resources available, staff training, and patient adherence.
    • This has now been covered in call for consensus statement (discussion, conclusion), but we did not elaborate on the underlying challenges as this would have required detailed description of workforce in UK vs Europe.
    • Further, a more detailed discussion is warranted about the discrepancies between guidelines and practical implications
      • The authors have renamed consensus vs discrepancies to similarities vs differences and have described the implications from the review.
    • Areas for Further Investigation:Provide a list of areas where additional research is required, particularly if they are items with disagreement in the guidelines. This will inform future studies and potential guideline revision.
      • This has been included in the discussion for NICE/ISWP, as well as the one broad relevant recommendation from ESO.

Conclusion:

  • Reinforce the argument in the conclusion and the fact that unifying the guidelines could lead to enhanced patient care and outcome.
    • A consensus statement has now been added
    • Provide proposed recommendations on harmonizing the guidelines or covering any gaps identified, developing a more consistent upper limb rehabilitation post-stroke approach.
      • A consensus statement has now been added
      • Research recommendations are included in the description

Reviewer 2 Report

Comments and Suggestions for Authors

Congratulations to the authors for the manuscript. The topic is relevant, and it is deeply needed for consistency in guidelines in stroke rehabilitation. The aim of this manuscript was to compare different guidelines published by organizations in the UK to be used as a decision-making guide in rehabilitation. The authors identified differences and similarities between the publications and provided an insight about gaps and future directions.

Title: Appropriate, concise and reflects the purpose of the manuscript

Abstract: The abstract provides a good idea of the study, but the results are not well described. I would suggest to add more specific information about your findings.

Introduction:

-        Line 40: This sentence is a little unclear. Are you saying that upper extremity impairment is present in 50% of the stroke survivors only at the 6 months mark or that they still have upper extremity impairment 6 months after the stroke?

-        Line 45: I would suggest some edits in the paragraph under the “persistent upper limb…” heading. When the sentence is too long, it hard to the reader to gather what is the point the author is trying to make. The information provided is also repetitive.

-        Line 59: It is not clear what is “external aspirations”. Please, clarify.

Materials and Methods

-        In my opinion, this section should be expanded. Details are needed to allow reproducibility of the methods and to understand the rigor that was applied in this process.

o   Did you use a system that was used in previous manuscripts?

o   What did you consider a consensus and what was a discrepancy?

o   How did you come up with the categories listed on Appendix 1?

o   What did you do in case of disagreement or uncertainty? Did you invite a third reviewer for that?

o   Did you read the guidelines individually and then compared the categories that each author identified?

o   How was data collected, organized, and analyzed?

-        I am not sure what type of information you are delivering on Appendix 1. Is that part of your results or your methods?

Results

-        Table 1:

o   strength training – you mentioned that there were no major discrepancies. Does that mean that there were minor discrepancies? How did you decide if it was major or minor discrepancy? This goes back to my comment about the methodology. All of that should be described. Also, some guidelines recommend considering and offering, but what is the difference? Is one recommending the application and the other is suggesting that it is optional? Is it related to level of evidence?

o   On the task-specific / RTT row: on the discrepancies column, on the first bullet point, you mentioned that the examples of RTT provided by NICE included “reading and grasping”. Is that correct? Did you mean reaching instead of reading?

-        Table 2:

o   On mental practice row: you mentioned that ISWP and ESO recommend and strongly recommend it. I believe that it can either be one or the other on each guideline. For example, ISWP recommend and ESO strongly recommend. Is that correct?

o   On the CIMT row: on the consensus column, you mentioned that the technique is “supported in some form…”, but you list the recommendations provided by all guidelines below that. It seems redundant to me.

o   Virtual reality row: on the consensus column, you mentioned that NICE and ISWP “avoid” mentioning VR. Did they avoid or not mentioned in the guidelines? And then you reported that both guidelines suggest that VR should be considered during rehabilitation. I think this row should be edit for clarification.

-        Table 3:

o   General recommendations row: in the consensus column, you added that NICE suggests considering splints, but you also mentioned that ISWP do not recommend splinting. Wouldn`t that be considered a discrepancy?

o   Electrical stimulation row – consensus column: you mentioned no major consensus, but is there any at all?

-        After the tables, you summarized the information using the headings and subheadings following the categories showed in the tables. It is a little confusing and the information is broken. I would suggest reorganizing the results to provide a take home message of each table.

Discussion

-        You provided a good summary of the main findings of your study, but it would be interesting to discuss the implementation of these recommendations, as well as the barriers to deliver these interventions in the clinical settings, access and applications of these guidelines by therapists and clinicians.

Conclusion

-        What are the take home message of your review? This information is missing here.

Comments on the Quality of English Language

There are some sections of the paper that need some edit for clarification. 

Author Response

Thank you very much for your feedback. Please see the authors' responses in the cover letter attached

Reviewer 3 Report

Comments and Suggestions for Authors

This study aimed to analyze the similarities and differences between the UK rehabilitation guidelines and the European Stroke Organization guidelines. The research is relevant, original, and offers a valuable contribution to the knowledge of the area. In that sense, I congratulate the authors. The manuscript, however, needs revision in some points. I present some suggestions and questions here for the proper application of the findings:

 Title: Adequate.

 Abstract: The abstract section is well done. I suggest the authors detail the conclusions more. After all, is it possible to affirm which guideline is best?

 Introduction: Since the authors restricted the analysis to the upper limb, they should detail more the physiological aspects affected by stroke, such as brain regions causing upper limb dysfunctions (pre-central area?), differences between cerebral arteries (anterior cerebral artery, middle cerebral artery, posterior cerebral artery), differences between plegia, paresis, and apraxia, and how ischemic and hemorrhagic stroke differently affect the upper limb.

 Materials and Methods: The authors should explore this topic more, explaining how comparisons were made. Besides, they should detail each topic of each instrument. In spite of this being a review, three rows are definitely not sufficient to detail such a topic. For example, the authors should explore why they are detailing the topics presented in the Results section.

Results: A couple of figures could make this section more attractive to readers.

 Discussion: This topic is well done. I congratulate the authors on providing a full discussion on the two instruments. I believe hat this study should state something about the American Stroke Association guidelines, which are as important as or more important (applicable) than the UK and European guidelines.

Conclusion: After all, is it possible to affirm which guideline is best? Which should we follow? These questions should be answered.

References: Adequate

Author Response

(The authors gave the same response as above.)

Round 2

Reviewer 1 Report

Comments and Suggestions for Authors

Authors have successfully addressed my comments.

Reviewer 3 Report

Comments and Suggestions for Authors

I congratulate the authors for addressing all the points. This version shows significant improvement compared to the initial version. My final suggestion is for the authors to format the study according to the journal's layout guidelines.